# Prostate-Specific Membrane Antigen Biology and Pathophysiology in Prostate Carcinoma, an Update: Potential Implications for Targeted Imaging and Therapy

**DOI:** 10.3390/ijms25179755

**Published:** 2024-09-09

**Authors:** Justine Maes, Simon Gesquière, Anton De Spiegeleer, Alex Maes, Christophe Van de Wiele

**Affiliations:** 1AZ Groeninge, 8500 Kortrijk, Belgium; 2Department of Diagnostic Sciences, University Ghent, De Pintelaan 185, 9000 Ghent, Belgium; 3Department of Geriatrics, University Hospital Ghent, 9000 Ghent, Belgium; 4Department of Morphology and Functional Imaging, University Leuven, 3000 Leuven, Belgium

**Keywords:** PSMA-pathophysiology-targeted, imaging and therapy

## Abstract

Prostate-specific membrane antigen (PSMA), a transmembrane glycoprotein, was shown to be expressed 100–1000 fold higher in prostate adenocarcinoma as compared to normal prostate epithelium. Given the enzymatic function of PSMA with the presence of an internalization triggering motif, various Glu-urea-Lys-based inhibitors have been developed and, amongst others, radiolabeled with positron emitters for targeted positron emission tomography imaging such as ^68^Ga-PSMA-HBED-CC Glu-urea-Lys(Ahx) as well as with beta and alpha-emitting radioisotopes for targeted therapy, e.g., ^177^Lu-PSMA-617. In this paper, we review and discuss the potential implications for targeted imaging and therapy of altered PSMA-glycosylation, of PSMA-driven activation of the P13K/Akt/mTOR, of the evolution over time and the relationship with androgen signaling and changes in DNA methylation of PSMA, and of androgen deprivation therapy (ADT) in prostate carcinoma.

## 1. Introduction

Prostate carcinoma is the most common cancer and the most frequent cause of cancer-related death in men [1]. Radiotherapy and radical prostatectomy with curative intent are the mainstays of locally confined prostate carcinoma [2,3,4]. Locally advanced and metastatic prostate carcinoma is treated palliatively, primarily through androgen deprivation therapy (ADT), given prostate carcinoma proliferation is highly dependent on androgens. Unfortunately, patients under ADT will ultimately progress as a result of castration resistance. Over the past years, the treatment armamentarium of castration-resistant prostate carcinoma has significantly increased. Currently approved drugs for treating mCRPC include androgen-pathway targeting drugs such as abiraterone and enzalutamide, chemotherapeutics such as docetaxel and cabazitaxel, autologous vaccination (sipuleucel-T), [^223^Ra]-dichloride and more recently [^177^Lu]-PSMA-617 which targets the prostate-specific membrane antigen (PSMA) [5].

Prostate-specific membrane antigen is a transmembrane glycoprotein highly overexpressed in prostate carcinoma, first described in 1983 by Horoszewicz et al. using LNCaP prostate carcinoma cells for immunization in a search for the identification of a prostate-specific antigen that could be used for immunohistochemical identification of prostate cancer cells [5]. PSMA was shown to be expressed 100–1000 fold higher in prostate adenocarcinoma as compared to normal prostate epithelium [6]. Given the enzymatic function of PSMA with the presence of an internalization triggering motif, various Glu-urea-Lys-based inhibitors have been developed and, amongst others, radiolabeled with positron emitters such as [^68^Ga]-PSMA-HBED-CC Glu-urea-Lys(Ahx) for positron emission tomography (PET)-imaging as well as with beta and alpha-emitting radioisotopes for therapy, e.g., [^177^Lu]-PSMA-617 [7]. The latter agents are currently part of the routine diagnostic and therapeutic armamentarium of prostate cancer patients.

In this paper, we review the currently available literature on the role of PSMA in the genesis and progression of prostate carcinoma, as well as the impact of ADT on its expression. The search was performed on Pubmed using the search terms “prostate-specific membrane antigen”, “PSMA”, ”FOLH1”, “glutamate carboxypeptidase II”, or “folate hydrolase” in conjunction with “prostate cancer” or “prostate carcinoma”, as well as one of the following terms: “biology”, “expression”, “progression”, “signaling”, “epigenetic”, “androgen”, “folate”, “homodimer”, or “dimerization”. Potential implications for the optimization of PSMA-targeted imaging and therapy are discussed.

## 2. Prostate Specific Membrane Antigen (PSMA)

### 2.1. Structure

PSMA, also termed FOLH1 or glutamate carboxypeptidase 2, is a type 2 integral membrane glycoprotein possessing a 19-aa cytoplasmic fragment, a single 24-aa membrane-spanning cytoplasmic fragment, and a 707-aa extracellular region [8]. It is natively expressed as a noncovalent homodimer but can form dimers, which appears to be important for immunization, given only dimers are able to elicit antibodies that efficiently recognize PSMA-expressing tumor cells [9]. For dimerization, the extracellular region appears to be sufficient given a truncated PSMA protein lacking the transmembrane, and the cytoplasmic domain was shown to also form dimers. Helical dimerization occurs between residues 601–750 of the extracellular region. Although of moderate affinity in solution, the monomer-dimer equilibrium may be sufficient to maintain high-level dimerization in the more restricted two-dimensional space of the cell surface, with the transmembrane and cytoplasmic region likely contributing to dimer stabilization.

### 2.2. Enzymatic Function

While PSMA is present at the level of the cell membrane either as a monomer or dimer, dimerization is a prerequisite for its enzymatical activity [9]. PSMA is capable of cleaving terminal glutamate from the neurodipeptide, N-acetyl-aspartyl-glutamate (NAAG), present in neuronal synapses (Naaladase activity) [10,11]. Furthermore, it can catalyze the cleavage of terminal glutamates from poly- and gamma-glutamate folates, present in dietary components at the level of the small intestine, thus generating folates that can subsequently be taken up by cells primarily via the reduced folate carrier. The substrate binding cavity in its C-domain contains a patch of three arginines (Arg-463, -534, and -536) that are clustered within 4.5 Ä of each other and located 6–12 Ä from the nearest zinc ion. In addition, a fourth Arg, Arg-210, located in the apical domain, is situated on the opposite side of the cavity and is likely also involved in substrate binding [10]. Conservation of all four arginine residues was shown to be essential for the ability of PSMA to cleave NAAG, consistent with the proposed role of these arginine residues in binding substrates containing C-terminal glutamates. Glutamate residues of PSMA substrates are likely to form electrostatic interactions with the arginine patch of PSMA, with the C-terminus binding to zinc ions. As shown by Gosh et al. and Barinka et al., aside from the conservation of the arginine residues, glycosylation of PSMA is of paramount importance for its enzymatic function given that partial or complete removal of sugar residues, either enzymatically or via mutagenesis, was shown to reduce or abolish its enzymatic function [12,13]. The extracellular portion of PSMA contains some 10 potential N-linked glycosylation sites. Glycosylation is also key to the targeting of the protein to the cell membrane, to proper protein folding, and to its residence rate at the cell membrane with the removal of sugars, resulting in an increased rate of degradation.

### 2.3. Internalization of PSMA and Routing to the Endocytic Compartment

PSMA undergoes spontaneous, constitutive internalization, which may be enhanced in a dose-dependent manner by PSMA-targeting specific antibodies [14]. Both in vitro studies and in silico modeling approaches have shown that ligand-bound PSMA is internalized about five times greater than unbound PSMA. Internalization rates of antibody-bound PSMA have been described as 60% within 20 min of saturation, compared to two hours for 60% unbound PSMA. Internalization is mediated via clathrine-coated pits followed by trafficking through the endocytic compartment [15]. Following routing to the endocytic compartment, the PSMA molecule can either be degraded or recycled back to the cell membrane. As shown by Rajasekaran et al., its internalization is mediated by the five N-terminal amino acids (MWNLL) present in its cytoplasmatic tail; deletion of the latter five amino acids abolishes its internalization [16]. More specifically, the first amino acid M (methionine) and the last L (leucine) appear essential for internalization, and the MXXXL signal was further proven to be a lysosomal routing signal. For recycling to the endocytic compartment, the association of the cytoplasmic tail of PSMA with Filamin a (FLNa) is mandatory, given that abolishing expression of the latter protein results in cytoplasmic accumulation of the internalized PSMA [17]. 

### 2.4. Regulation of PSMA Expression and PSMA Splice Variants

The PSMA-gene (FOLH1) was shown to be located on chromosome 11p11.2 by O. Keefe et al. and shown to consist of 19 exons spanning approximately 60 kB of genomic DNA [18]. As opposed to basal transcription via the proximal 1.2 kb promotor/lead region, transcription was shown to be enhanced 40-to 144-fold via a 90 bp enhancer located in the third intron, about 12 kB downstream from the start of the site of transcription. This enhancer was shown to be characterized by a 72 bp direct repeat within a 331 bp core region as well as to activate transcription from its own as well as of heterologous promoters in prostate cancer cell lines [19]. While transcription of the gene via the promotor/lead region is not influenced by androgens, transcription via the enhancer is significantly downregulated by androgens [20]. Four AR-binding peaks involving introns of the PSMA (FOLH1) gene have been recorded, as well as direct repeat regions harboring nine copies of SRY/SOX sites, SRY, SOX7, and SOX18, suggesting that SRY or SOX may interact with the androgen receptor binding domain, and as a result, AR may sequester these tissue-specific proteins, causing repression of the enhancer [21]. The enhancer region was also shown to have eight API binding sites at the repeat region. More recently, Bakht et al. identified the homeobox transcription factor HOXB13, NKX3, and the forkhead box1I protein (FOXA1) as positive regulators of the PSMA-enhancer, regardless of AR-status, with knockout models pointing to HOXB13 as an upstream regulator of PSMA in both AR-positive and AR-negative prostate carcinoma [22]. Their data suggest that while AR and HOXB13 cooperate to regulate PSMA via enhancer binding in the setting of AR-positive disease, if AR expression is lost, HOXB13 may continue to regulate PSMA on its own. Finally, a CpG island at the 5’ terminus and a 14kB region encompassing the first 5 introns of the FOLH1 gene have been identified, the cytosines of which can be methylated by DNA-methyltransferases, resulting in gene silencing [23]. Treatment with histone deacetylase inhibitors was shown to reverse such repression and restore PSMA expression both in vitro and in vivo. As for splice variants of PSMA, four have been identified, respectively: PSM’, PSM-C, PSM-D, and PSM-E. PSM’ lacks the intracellular and transmembrane domains, resulting in a cytosolic protein that bears enzymatic activity as a dimer [24]. The splice variant PSM-C generates a protein identical to PSM’. PSMA-D, which has a new translation initiation start in exon 1C, followed by 42 novel amino acids and the rest of the PSMA protein frame. PSMA-E has a similar exon insertion and bears a 93bp deletion at nucleotides 2223–2324.

## 3. PSMA in Prostate Carcinoma

### 3.1. Pathophysiology

Preclinical data: Folate is a B-vitamin present in cells as a family of co-enzyme factors that carry and chemically activate 1-carbons that are required for “de novo” synthesis of purines and thymidines and for the methylation of homocysteine to methionine [25] and thus for cell proliferation. In a study by Yao et al., folate uptake and proliferation of PC-3 prostate carcinoma cells grown in the presence of poly-gamma-glutamated folate proved significantly higher than that of PC-3 vector cells. Both uptake and proliferation were attenuated by the addition of the specific PSMA-inhibitor 2-(phosphonomethyl)-pentanedioic acid (2-PMPA) [26]. The authors also found that PSMA-expressing cells demonstrated a 2-fold increase in uptake of tritiated folate, suggesting that PSMA may also be associated with folate transport. In in vivo animal studies, tumors expressing PSMA (LNCaP) had a significantly higher take rate (16/18) compared to tumors not expressing PSMA (shLNACP) (18/18). LNCap PSMA-expressing tumor cells cooperated with both the proton-coupled folate transporter and the reduced folate carrier to increase folate uptake [27]. Additionally, PSMA-expressing tumors grew significantly larger, and this was further increased in the presence of dietary folate. Inversely, treatment with 2-PMPA did not affect the growth of xenografts that were lacking PSMA, indicating that the inhibitor’s therapeutic effect is associated with inhibition of the enzymatic activity of tumor-expressed PSMA. Caromile et al. observed that in the transgenic adenocarcinoma of the mouse prostate (TRAMP) murine model, primary prostate carcinoma was larger in wild-type animals when compared to PSMA knockout animals and was highly vascularized with higher rates of progression from hyperplasia to adenocarcinoma. The authors found that PSMA-positive tumor cells were able to survive at greater distances from the vasculature with less cell death than PSMA knockout cells [28]. When present in a protein complex with the cell scaffolding protein receptor for activated C kinases (RACK1), signaling between beta1-integrin and IGF-1R was disrupted, resulting in activation of the P13K-AKT pathway instead of the canonical MAPK-ERK1/2 pathway. Using gene set enrichment analysis, Kaittanis et al. demonstrated that in prostate carcinoma cells expressing PSMA (LNCaP-Ctrl and PC3-PSMA) but not in prostate carcinoma cells that lacked PSMA (LNCaP-KD and PC3-Ctrl), the levels of genes involved and regulated by Akt and mTOR were elevated and that PSMA expression increased the phosphorylation of Akt and its downstream targets S6K and 4EBP1, indicating the activity of Akt-mTOR is modulated via PSMA. The authors further demonstrated that PSMA initiates signaling upstream of P13K via the metabotropic glutamate receptor through its release of glutamate from Vit B9 and other glutamate substrates that activate mGLur1 [29]. The activated mGluR I subsequently induces activation of phosphoinositide 3-kinases (PI3K), known to play a particularly important role in the pathogenesis of prostate cancer, through phosphorylation of p110 beta independent of PTEN (the Phosphatase and Tensin Homolog suppressor gene) loss. Hong et al. constructed PSMA-knockdown LNCaP and 22rv1 cell lines and, performing metabonomic and transcriptomic analyses, found that PSMA knockdown decreased the expression of the metalloproteinase MMP-7, affected the extracellular matrix disassembly and organization, and promoted the biosynthesis of arginine and proline. MMP-7 has been previously shown to promote PCa-induced osteolysis via solubilization of the TNF family member receptor activator of the nuclear kappa-beta ligand (RANKL) [30]. Of interest, knockdown also resulted in the activation of AR signaling. 

Data in patients (see Table 1): Sweat et al. studied 232 patients with node-positive prostate adenocarcinoma who underwent bilateral lymphadenectomy and radical prostatectomy and, using IHC, found PSMA expression in all primary tumors and 98% of involved lymph nodes. The intensity of staining proved greatest in primary carcinoma [31]. Mannweiler et al. studied 51 patients with primary prostate carcinoma and distant metastases using IHC. Out of the 51 primary lesions, 7 presented a heterogeneous expression, and 2 proved negative, whereas out of the 51 distant lesions, 6 presented a heterogeneous expression, and 8 were negative [32]. Sayar et al. performed IHC on 636 samples from 339 anatomically distinct metastatic sites of 52 cases of the UW-TAN cohort and found a trend towards lower PSMA levels in AR- as compared to AR+ tumors [23]. PSMA levels appeared to follow a bimodal distribution, with the largest groups of samples showing either a low or high expression. In addition to intertumoral heterogeneity, the authors also noted substantial intra-tumoral differences in PSMA expression, with PSMA-high and PSMA-negative cell populations co-existing within a given metastasis, as well as PSMA expression on the neovasculature of some PSMA-prostate lesions. Using different sets of patient population samples from the LuCa PDX, UW-TAN, and SU2C databases and gene set enrichment analysis, PSMA low/negative tumors were shown to show strong enrichment of genes involved in cell proliferation, inflammatory response, hypoxia, and metabolic and glycolytic pathways. Also, an inverse relationship between MUC1 and PSMA expression was identified, and loss of PSMA expression proved associated with gain of CpG methylation and loss of histone 3 lysine 27 acetylation. Treatment with histone deacetylase (HDAC) inhibitors was able to reverse this epigenetic repression and restore PSMA expression in vitro and in vivo in cell lines and animal models. Finally, in AR+ samples, AR binding at the FOLH1 locus and AR signaling activity did not differ significantly between PSMA-high and PSMA-low levels. Bakht et al. performed transcriptome analysis on data from the SUC2 and PCF databases and identified overall lower PSMA gene expression on liver metastases when compared to other sites of metastatic disease [22]. PSMA suppression was associated with promotor histone three-lysine methylation and higher levels of neutral amino acid transporters. A subset of AR-negative CRPC with high PSMA was identified with models pointing towards HOXB13 as an upstream regulator of PSMA in both AR+ and AR- CRPC. In a study by Paschalis et al., PSMA expression levels proved to be nearly three times as high in mCRPC versus CSPC [33]. In their study, 16 out of 38 CSPC tissue samples and 16 out of 59 mCRPC biopsies proved PSMA negative. PSMA expression demonstrated marked inter- and intra-patient heterogeneity. Kaittanis et al. analyzed genomic data from prostate cancer patients and found that higher PSMA expression was positively related to faster biochemical recurrence (higher PSMA expression was defined as a z-score >2 and low PSMA expression as a z-score ≤ 2) and metastasis [29]. At 160-month follow-up, approximately 70% of patients with low PSMA were still disease-free, whereas the vast majority of patients with high PSMA expression had relapsed (Kaplan–Meier log-rank, *p* = 0.039). Likewise, over 80% of low-expression PSMA patients were metastasis-free at 250-month follow-up, whereas the majority of high-expression PSMA patients had developed metastasis (log-rank test, *p* = 0.019). Using immunohistochemistry and gene set enrichment analysis, the authors subsequently demonstrated that increased PSMA levels in their patients were associated with phosphorylation of 4EBPI, a downstream target of mTOR, and at the transcriptional level with changes in transcription of mTOR-regulated genes. Tissue microarray analysis of samples from 76 prostate cancer patients further showed that expression of PSMA correlates with phosphorylation of the Akt target 4EBP1. Using [^68^Ga]-PSMA-HBED-CC PET imaging and magnetic resonance imaging before prostatectomy and quantitative immunofluorescence microscopy for PSMA and phosphorylated Akt, principal component analysis identified that PSMA expression and tracer uptake were more strongly correlated with Akt’s phosphorylation at S473 than other clinical indicators of prostate cancer, such as Gleason score and PSA, suggesting that PSMA-based PET imaging can serve as a predictor of the global activation status of the P1K-Akt pathway in prostatic lesions. Finally, using RNAse protection assays, Su et al. examined the expression of PSMA and PSM in normal, hyperplastic, and cancerous prostate tissue. PSM/PSA ratios ranged from 3 to 6 in prostatic carcinoma tissue, from 0.75 to 1.6 in benign hyperplastic prostate tissue, and from 0.075 to 0.45 in normal prostate tissue [34]. Comparable results were reported by Schmittgen et al. in a larger patient population [35].

### 3.2. PSMA and ADT in Prostate Carcinoma

Preclinical studies: Enhanced PSMA expression on PSMA-positive prostate carcinoma cell lines xenografts upon treatment with enzalutamide and androgen deprivation has been reported by several authors [36]. In a series by Lückerath et al., pre-treatment with enzalutamide before RLT led to more substantial DNA damage when compared to RLT monotherapy but did not result in additional growth retardation [37]. Kranzbühler et al. treated LNCaP prostate cancer cells and PNT1A epithelial prostate cells with enzalutamide, dutasteride, rapamycin, metformin, Lovastatin, and acetylsalicylic acid and assessed its impact on PSMA and AR expression using flow cytometry, immunohistochemistry, and immunoblotting. Furthermore, the authors also performed [^177^Lu]-PSMA-617 uptake and internalization studies [38]. Enzalutamide and dutasteride were shown to result in a significant upregulation of PSMA surface levels in LNCaP cells, whereas total PSMA expression proved to be significantly enhanced after treatment with enzalutamide and rapamycin. Uptake and internalization of [^177^Lu]-PSMA-617 were both significantly increased in LNCaP cells following treatment with enzalutamide and rapamycin. Staniszewska et al. studied the effect of enzalutamide on PSMA expression over a 3-week period in the PSMA-low prostate cancer cell line 22Rv1 and the PSMA-high prostate cancer cell lines C4-2 and LNCaP [39]. Treatment was shown to result in an increase in PSMA expression in all three cell lines already after one week of treatment. After two weeks of treatment, follow-up [^68^Ga]-PSMA-11 PET/CT images were performed and compared to baseline PET/CT images in xenografts of the same cell lines grown in nude mice. Uptake/gram tissue in 22Rv1 tumors was shown to increase significantly after two weeks. Liu et al. studied the long-term effect (20 passages) of androgen deprivation on AR and PSMA expression in the androgen-sensitive LNCaP cells and the AR and PSMA-negative PC-3 cells, which was used as a negative control [40]. After 10 passages, AR-protein expression and PSMA expression were no longer detectable, suggesting their expression is dependent on the androgen levels and length of time of androgen deprivation during cell growth. Of interest, analysis of PSMA relative enzymatic activity revealed that there was an apparent increase for whole-cell protein samples for 2 days and 5 passages when compared to normally cultured LNCaP cells, while labeling with a fluorescent inhibitor decreased for these time points. These findings suggest a possible post-translational modification, such as a change in N-glycosylation pattern with improved enzymatic activity and a relative decrease in cell surface PSMA expression or loss of inhibitor affinity. In the PSMA and AR-negative cell lines DU145 and PC3, the loss of expression of both was found to be due to epigenetic silencing by CpG island hypermethylation of their promotor regions. Sommer et al. studied the effects of manipulating AR activity on PSMA expression in two castration-sensitive PCa cell lines with non-detectable PSA protein level (LAPC4) and average PSMA protein level (LNCaP), as well as in the LNCaP sub-cell line C4-2, representing a castration-resistant cell model with high PSMA expression [41]. In line with previous findings, incubating these cell lines with the synthetic androgen R1881 resulted in a concentration-dependent decrease in FOLH1 mRNA and PSMA protein levels, an effect that could be partially reversed by antiandrogen treatment. Four different anti-estrogens were assessed for their potential to reverse the effect of R1881, respectively: bicalutamide, enzalutamide, apalutamide, and darolutamide. In LAPC4, all tested antiandrogens reversed the inhibitory effect of R1881, whereas in LNCaP, only apalutamide treatment resulted in a significant reversal of the inhibitory effect of R1881. IN C4-2 cells, all antiandrogens, except for enzalutamide, partially abolished the inhibitory effect of R1881. Additional analysis of the publicly available AR chromatin immunoprecipitating sequencing dataset for the loci of FOLH1 in the AR-positive cell lines LNCaP, C4-2, and 22Rv1 revealed that the AR is not mandatory for FOLH1 gene expression given canonical ARB could not be found.

Clinical studies, IHC: Wright et al. performed immuno-peroxidase staining for PSMA reactivity on prostate carcinoma specimens from 20 prostate carcinoma patients who matched pre-treatment and post-treatment (treatment consisting of medical or surgical castration or combination androgen deprivation therapy); specimens were available as well on specimens from 16 patients that were obtained post-treatment (5 from primary and 11 from metastatic lesions) [42]. PSMA reactivity was found to be increased in 11 of 20 post-treatment primary tissues and in all of the post-treatment metastatic specimens. Sommer et al. assessed PSMA expression using IHC in prostate carcinoma tissue before and during androgen deprivation therapy in 35 treatment-naïve patients, 55 tissue samples of patients under ADT, and 15 tissue samples from patients under ADT + NHT [41]. ADT was induced by treatment with buserlin, triptorelin, degarelix, or leuprorelin for at least one month. For ADT + NHT, ADT was combined with enzalmutamide or arbiraterone. No significant difference was found between treatment-naïve HSPC and ADT/ADT + NTH. To assess if further treatment with antiandrogens impacted PSMA expression, ADT and ADT+NHT patients were analyzed separately. It was found that ADT+NHT marginally but insignificantly increased PSMA expression. The authors also analyzed the TURP tissue of 6 patients before and under ADT and found PSMA to be significantly increased in 3 out of 6 patients. Closer analysis of their data showed, however, that patients with a low PSMA before and during ADT presented with an increase in PSMA expression, whereas patients with a high PSMA expression presented with a stable or decreased PSAM expression under ADT.

Clinical studies, PET imaging (see Table 2): Afshar-Oromieh retrospectively studied the effect of continued ADT treatment, range 42–369 days, on PSMA expression using [^68^Ga]-PSMA-11 PET/CT in ten patients. Of the 31 lesions visible on the pre-ADT examination, only 14 lesions were visible in eight of the 10 patients on the post-ADT examination [43]. Furthermore, of all 31 lesions, one-third were still visible in 6 patients with a complete PSA response (< 0.1 ng/mL). Hoberück et al. retrospectively studied 21 therapy-naïve patients who underwent [^68^Ga]-PSMA-11 PET/CT or PET/MRI prior to as well as after initiation of ADT (range 61–289 days) [44]. On follow-up, in all patients, a relevant decrease in lesion count occurred, and in those patients with a residual PSA < 1 ng/mL, 49 known lesions were no longer visible on the post-ADT. Gupta et al. obtained [^68^Ga]-PSMA-11 PET/CT scans at baseline and under ADT (range 3–12 months) in 43 patients and found that after a median treatment of 6 months, 23.3% of nodal sites of involvement and 17.6% of metastases were no longer visible [45]. Reduction in PSMA uptake proved higher in M0 when compared to M1 disease. Emmett et al. performed serial [^68^Ga]-PSMA-11 PET/CT scans at baseline and on days 9, 18, and 28 in 8 men with hormone-sensitive PCa commencing LHRH +/− bicalutamide and 7 men with castration-resistant PCa commencing NHT (enzalutamide or arbiraterone) [46]. Meanwhile, in the hormone-sensitive group, the intensity of uptake progressively decreased in the majority of patients, while in the CRCP group, SUV-max progressively increased. Ettala et al. prospectively studied nine treatment-naïve prostate cancer patients prior to 1.5 weeks (range 0.8–2.5 weeks), 2.9 (1.9–4.5 weeks), and 6.2 weeks (3.5–8.7 weeks) following initiation of ADT [47]. Four patients presented with local or locally advanced disease, while the remaining four suffered from distant metastatic disease. All patients reached castration levels within 10 days, and a 50% decrease in PSA levels was observed 14 days post-ADT. In one patient, three new bone metastases were observed post-ADT. The increase in [^68^Ga]-PSMA-11 uptake was heterogeneous and most evident in bone metastases in the remaining seven patients. In two patients, no change in the intensity of tracer uptake was found. In a more recent study by the same group, 25 newly diagnosed, treatment-naïve metastatic prostate carcinoma patients prospectively underwent [^18^F]-PSMA-1007 PET/CT imaging before and 3–4 weeks after ADT initiation [48]. All patients reached castration levels of testosterone at the time of the second scan. Following ADT, 104 of 404 bone lesions, 33 out of 314 lymph node lesions, and 6 out of 57 primary prostate carcinoma lesions showed an increase (≥20%) in tracer uptake. In 22 out of 23 patients with bone metastases, a flare phenomenon was observed, and in 10 out of these patients, new tracer-avid lesions were identified. Patients also underwent an [^18^F]-FDG (fluorodeoxyglucose) PET/CT baseline examination, and those lesions that presented a flare had a less intense FDG uptake.

## 4. Discussion

### 4.1. Altered PSMA-Glycosylation on Prostate Carcinoma, Potential for Optimization of RLT Efficacy

The spontaneous, constitutive internalization of PSMA was shown to be enhanced in a dose-dependent manner using the PSMA-targeting specific antibody J591 [14]. Internalization rates of antibody-bound PSMA were estimated at 60% within 20 min of saturation, compared to two hours for 60% unbound PSMA. The therapeutic potential of [^177^Lu]-J591 in mCRPC has been previously evaluated in phase I and II trials, with response rates defined as any PSA response varying from 33 to 60% at doses ranging from 10 to 70 mCi/m2 [49]. When comparing the biodistribution of [^68^Ga]-PSMA-HBED-CC and [^89^Zr]-J591, an absent uptake of tracer in the salivary glands as well as in the small bowel is seen on the [^89^Zr]-J591 images [50]. Given the therapeutic efficacy of [^177^Lu]-J591, this finding shows that J591 targets an epitope on the extracellular domain of PSMA solely available on PSMA exposed to prostate cancer cells and not on healthy cells. It is estimated that approximately 25% of the molecular weight of PSMA is made up of glycosyl chains, either N-linked or O-linked, and N-glycosylation, particularly at distal sites. It has become well accepted that altered glycosylation is a hallmark of cancer progression, including mCRPC, and complex-type glycans lacking polylactosamine have been observed on PSMA derived from prostate cancer tissue and serum, while primarily high-mannose forms have been reported in the LNCaP setting [51]. Thus, the specificity of J591 for prostate carcinoma tissue is more than likely related to specific changes in the glycosylation of PSMA. Hypothetically, cold J591 or derivatives with a smaller molecular weight co-injected with [^177^Lu]-PSMA-617 may increase the internalization rate of [^177^Lu]-PMSA and [^68^Ga]-PSMA-HBEDC in prostate carcinoma tissue, thereby reducing uptake in healthy tissue, e.g., the salivary glands. If so, this would allow for an increase in diagnostic and therapeutic efficacy while reducing therapeutic toxicity, predominantly pertaining to the salivary glands. 

### 4.2. Evidence Linking Folates to the Development and Growth of Prostate Carcinoma

In a double-blind, randomized trial including 634 men who were randomly assigned to folate supplementation or aspirin for the prevention of colorectal adenoma, a total of 34 subjects were diagnosed with prostate cancer after random assignment over a period of 10 years [52]. The estimated probability of being diagnosed with prostate cancer was 9.7% in the folic acid group and 3.3% in the placebo group (age-adjusted hazard ratio 2.63). This study highlights the importance of folates in the development of prostate cancer. Folate is essential for the “de novo” synthesis of purines and thymidines for the methylation of homocysteine to methionine and, accordingly, for cell proliferation in general. In vivo, serum folate is non-poly-gamma-glutamated, ready for transport into tissues, and taken up by cells primarily via the reduced folate receptor. However, as shown by Yao et al., PSMA is more than likely also involved in direct folate uptake, given its presence in prostate carcinoma cells, resulting in a twice as high uptake of tritiated folate when compared to PSMA-negative cells [26]. Furthermore, PSMA can release folate from poly-gamma-glutamated folates through its enzymatic activity. However, contrary to brush border cells where PSMA is present at the basocellular level, in normal prostate tissue as well as in prostate carcinoma, PSMA is expressed at the apical surface. Thus, its expression at the apical surface suggests exposure to poly-gamma-glutamated folate and likely also free folate at this surface, possibly from dead or dying prostate carcinoma cells that release their content, including their stored poly-gamma glutamate folates in their surrounding environment where they become available to PSMA overexpressed on viable cancer cells. In vitro, prostate carcinoma cells expressing PSMA grown in the presence of polyglutamate folate were shown to have a significantly higher proliferation rate when compared to PSMA-negative cells, which could be attenuated by the PSMA-specific inhibitor 2-PMPA [25,26,27,28,29]. In various animal models, PSMA-expressing prostate carcinoma xenografts were shown to have a higher take rate, to be more vascularized, to grow larger, and to have a higher rate of progression when compared to non-PSMA-expressing prostate carcinoma xenografts, including knockouts. Furthermore, PSMA-positive tumor cells were shown to be able to survive at greater distances from the vasculature with less cell death when compared to PSMA knockout cells, suggesting that PSMA provides important cell-intrinsic survival components that also contribute to tumor growth. 

### 4.3. Glutamate Released from Gamma-Polyglutamated-Folates by PSMA, Not Such an Innocent Bystander

Aside from folates, in vitro, glutamates released from gamma-polyglutamated -folates by PSMA have proven responsible for the activation of the P13K-Akt-mTOR pathway, known to play an important role in cancer cell survival, angiogenesis, and metastasis via the metabotropic glutamate receptor. Using [^68^Ga]-PSMA-HBED-CC PET imaging and magnetic resonance imaging before prostatectomy and quantitative immunofluorescence microscopy for PSMA and phosphorylated Akt in prostate carcinoma patients, radioligand uptake proved most strongly correlated with Akt’s phosphorylation at S473, suggesting that PSMA-based PET imaging can serve as a predictor of the global activation status of the P13K-Akt-mTOR pathway in prostatic lesions [29]. Activation of the P13K/Akt/mTOR pathway has been reported to occur in approximately half of advanced prostate carcinoma with significant cross-talk between P13K/Akt/mTOR and AR signaling. Given the broad interest in the development of P13K inhibitors in cancer, PSMA-targeted imaging, e.g., [^68^Ga]-PMSA-HEBD-CC, may allow for the selection of those prostate carcinoma patients most likely to benefit from such novel therapeutic agents. 

### 4.4. PSMA Expression in Prostate Carcinoma, Its Evolution over Time and Relationship with Androgen Signaling and Changes in DNA Methylation

Based on IHC data, in patients suffering from prostate carcinoma, at the time of diagnosis, the majority of primary prostate cancer lesions and involved lymph nodes are PSMA positive, with the greatest intensity of staining being found in the primary lesions. As opposed to primary lesions and involved lymph nodes, distant metastases are more often present with a more heterogeneous PSMA expression, with high and low PSMA cell populations co-existing within given metastases and a significant proportion of metastatic lesions being PSMA [31,32,33,34,35]. Of interest, a trend towards lower staining in AR- metastases as opposed to AR+ metastases as well as PSMA-expression on neo-vasculature in AR- metastases has been reported. The latter finding may hypothetically, in part, explain the non-response of PSMA-avid lesions to [^177^Lu]-PSMA-617, given the latter is rapidly released from endothelial cells following internalization, which is not the case for prostate cancer cells [53]. While it was initially thought that PSMA expression is controlled largely by AR, more recent findings have identified HOXB13 as an upstream regulator for PSMA expression in both AR+ and AR- prostate cancer lesions. Overall, a lower to absent PSMA expression was reported in liver metastases from prostate carcinoma, and an overall low or absent level of PSMA in prostate carcinoma lesions was reported to be related to promotor histone 3 lysine methylation, gain of CpG methylation, and loss of histone 3 lysine 27 acetylation. In vitro and in vivo animal models suggest that treatment with histone deacetylase inhibitors may reverse these epigenetic changes in prostate cancer cells and restore PSMA expression, which in vivo could be readily monitored using PSMA-targeted PET/CT imaging.

### 4.5. Imaging Options in PSMA Low/Negative Prostate Carcinoma

Importantly, PSMA low/negative prostate carcinoma lesions have been shown to show strong enrichment of genes involved in cell proliferation, inflammatory response, hypoxia upregulation, amino acid transporters, and genes involved in glycolysis. The latter finding is in keeping with imaging studies using combined [^18^F]-FDG PET/CT and PSMA-targeted PET in mCRPC patients with a reported incidence of FDG+/PSMA- lesions ranging from 19 to 33% [54]. The presence of FDG+/PSMA- lesions in patients suffering from prostate carcinoma has been confirmed as a negative prognostic factor associated with a poorer response to [^177^Lu]-PSMA-617 treatment. The upregulation of amino acid transporters in PSMA low/negative prostate carcinoma further suggests that PET imaging with radiolabeled amino acids, e.g., [^18^F]-Fluciclovine PET/CT may provide comparable information to that of FDG-PET and also explains why [^18^F]-Fluciclovine PET/CT imaging is less performing for detection of recurrent prostate carcinoma when compared to PSMA-targeted PET-imaging [55]. Of interest, an inverse relationship between PSMA and MUC1 expression has also been reported. MUC1 has been shown to play an important role in malignant transformation, cell proliferation, cell renewal, and metastatic invasion. Accordingly, various therapeutic strategies targeting MUC1 have been developed, including immunotherapy, and their therapeutic efficacy has been demonstrated in vitro and in vivo models [56]. Hypothetically, FDG+/PSMA- findings in prostate carcinoma patients could serve as a surrogate for MUC1 expression and, hence, for the selection of mCRPC patients that might benefit from combined MUC1 and PSMA targeted therapy.

### 4.6. Impact of ADT on PSMA Expression in Prostate Carcinoma and Implications for PSMA-Targeted Imaging and RLT

Available studies looking at the impact of ADT on PSMA expression suggest that changes in PSMA expression following ADT are time-dependent, with an initial increase during the first couple of weeks in some patients accompanied by a flare phenomenon, followed by a decrease after long-term ADT [36,37,38,39,40,41,42,43,44,45,46,47,48]. The increase following short-term ADT is heterogeneous among lesions and most evident in bone metastases, with the appearance of previously unknown bone metastases in some patients. Given the well-documented relationship between AR-signaling and PSMA expression, hypothetically, the heterogenous response identified may relate to differences in androgen sensitivity of the various lesions identified, with those showing a slight or no increase in PSMA expression and thus the least ADT-sensitive being the least AR-dependent and thus likely also the most aggressive lesions. Identification of those lesions that are more prone to progress under ADT may, hypothetically, allow for more optimized targeted treatment using, for instance, stereotactic radiotherapy. Alternatively, a modification in the glycosylation status of PSMA following ADT may also lead to an increased rate of internalization of PSMA and, thus, of PSMA-ligands on PET-images, resulting in higher uptake values (SUV-value) [14]. As opposed to short-term treatment, long-term treatment was shown to result in reduced PSMA expression/ligand uptake in the vast majority of primary lesions as well as metastatic lesions, with only half of the lesions remaining detectable under treatment. Lesions no longer visible during long-term ADT proved especially lymph nodal lesions and bone metastases. The reduction in the number of detectable lesions proved further significantly more pronounced in those patients exhibiting the greatest reduction in PSA values, which is not surprising given the expression of PSA is mainly regulated by the AR. Importantly, in spite of a complete PSA remission following ADT, in one study, up to one-third of lesions could still be depicted on PSMA-targeted PET imaging, with at least one lesion remaining visible under ADT in the majority of patients [43]. The disappearance of lesions following ADT is likely predominantly due to the induction of apoptosis of androgen-sensitive prostate carcinoma cells, as demonstrated first by Huggins in 1966 [57]. Inversely, in some patients, a limited number of lesions also exhibited an increase in PSMA uptake following long-term treatment with ADT. In these lesions, PSMA expression is more than likely no longer AR-dependent but regulated via, e.g., HOX13. Overall, the time-dependent changes in PSMA expression under ADT, as described above, may offer some opportunities to explore using PSMA-targeted imaging. In prostate carcinoma patients who present with a biochemical recurrence that is not under ADT and present with a PSA value < 0.5 ng/mL, the overall sensitivity for recurrence detection is approximately 50% [58]. In this patient population, it may hypothetically prove worthwhile to initiate short-term ADT prior to imaging, given it may increase the sensitivity of this imaging modality. In those patients that present with a recurrence on PSMA-targeted imaging and are candidates for ADT, a control scan after 3 months of ADT, regardless of the PSA value post-treatment, may identify the non-ADT-responding lesions that might be targeted using SBRT or might benefit from additional systemic therapies, e.g., enzalutamide or arbiraterone acetate, chemotherapy, or [^177^Lu]-PSMA-617 treatment. Finally, the documented evolution of prostate carcinoma lesions from PSMA+/AR-responsive to PSMA+/AR non-responsive followed by a PSMA-, aggressive phenotype, as evidenced by available studies, suggests the use of PSMA-targeted therapies should be upfront rather than at the end of the treatment chain, as is currently the case.

## Figures and Tables

**Table 1 ijms-25-09755-t001:** PSMA expression in prostate carcinoma.

Authors	Technique	Material Derived from	Results
Sweat et al. [31]	IHC	−232 pts, RP+lymphadenectomy	100% of ppc +, 98% of LN PSMA+
Mannweiler et al. [32]	IHC	−51 pts, ppc and distant lesions	49/51 ppc+, 43/51 distant lesions +
Sayar et al. [26]	IHC Gene set enrichment analysis	UW-TAN cohort), 636 samples, 339 distant metastatic sites), LuCa PDX, UW-TAN, and SU2C cohorts	- trend ↓ PSMA expression in AR- v. AR+, bimodal expression, intra-tumoral PSMA heterogeneity, PSMA expression on neovasculature - PSMA low/-lesions show ↑ glycolysis, hypoxia, inflammation, upregulation of MUC1, gain of CpG methylation and loss of histone3lysine27 acetylation
Bakht et al. [22]	- transcriptome analysis	- SUC2 and PCF cohorts	↓ PSMA expression in liver M+, associated with promotor histone 3 lysine methylation and ↑neutral amino acid transporters, a subset of AR-/PSMA++
Paschalis et al. [33]	- IHC	- 38CSPC tissue samples, 59 mCRPC biopsies (matched)	Ratio PSMA expression mCRPC/CSPC approximating 3, 16/38 CSPC and 16/59mCRPC PSMA-
Kaiitanis et al. [29]	genomic data analysis, IHC, and gene enrichment analysis, tissue microarray analysis	MSCKK cohort PMID 25024180 −6 pts	↑ PSMA expression is associated with faster biochemical recurrence and metastases changes in transcription of mTOR-regulated genes- Akt phosphorylation
Su et al. [34]	RNAse protection assays	normal/hyperplastic and cancerous prostate tissue	PSM/PSMA ratio’s; 0.075–045 in normal tissue, 0.75–1.6 in hyperplastic tissue, and 3–6 in cancerous tissue
Schmittgen et al. [35]	Quantitative PCR	normal/cancerous tissue (45 ppc/6 involvedLN/14 M+)	PSM’ ↑↑ in metastases

Legend: IHC = immunohistochemistry, PCR = polymerase chain reaction, ppc = primary prostate carcinoma, LN = lymph nodes, AR = androgen receptor, MUC1 = mucine-1, CSPC = castration sensitive prostate carcinoma, mCRPC = metastatic castration-resistant prostate carcinoma, ↑ = high or increased, ↓ = low or decreased, M+ = metastasis, RP = radical prostatectomie, ppc = primary prostate carcinoma, PSM’ = PSMA splice variant, PSMA = prostate-specific membrane antigen, pts = patients

**Table 2 ijms-25-09755-t002:** PET-imaging studies assessed the impact on PSMA expression of prostate carcinoma lesions under androgen deprivation therapy.

Authors	Study Type	Nb of Patients	Treatment Duration, Range	Results
Afshar Oromieh et al. [43]	retrospective	10	42–396 d	14/31 lesions + in 8 pts, 1/3rd of lesions still + in 6 pts with a complete PSA response
Höberuck et al. [44]	retrospective	21	61–289 d	↓in all pts, in pts with post-therapy PSA <1 ng/mL, 49 lesions no longer visible
Gupta et al. [45]	retrospective	43	3–12 m	After a median treatment duration of 6 months, 23.3% of LN and 17.6% of M+ were no longer visible, ↓ in the PSMA signal ↑ in M0 vs. M1 disease
Emmett et al. [46]	prospective	15	28 d (imaging on d9, d18 and d28 post-treatment	In 8 hormone-sensitive patients, uptake progressively ↓ (LHRH +/− bicalutamide) In the 7 castration-resistant pts, uptake progressively ↑ (enzalutamide or arbiraterone))
Ettala et al. [47]	prospective	9	6.2 w (imaging at 1.5 w, 2.9 w and 6.2 weeks post-treatment)	In 1 pt, 3 new bone M+ were identified; in 7 pts, lesion count ↑; in 2 pts, no change was found
Malaspina et al. [48]	prospective	25	3–4 w	104/404 bone M+,33/314 LN and 6/57 ppc showed an ↑ in tracer uptake, in 22/23 pts with bone M+ a flare phenomenon was found

Legend. All patients underwent a baseline PSMA-targeted PET examination, treatment duration = timing, after which follow-up PET scans were performed. Study references are indicated between brackets. Pts = patients; ppc = primary prostate carcinoma; LN = lymph node; M+ = metastasis; LHRH = luteinizing hormone-releasing hormone; ↑ = high(er) or increase; ↓ = low(er) or decrease; nb = number of.

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
