# Peer review of "Prostate-Specific Membrane Antigen Biology and Pathophysiology in Prostate Carcinoma, an Update: Potential Implications for Targeted Imaging and Therapy"

_ijms, 2024, doi:10.3390/ijms25179755_

Round 1
Reviewer 1 Report
Comments and Suggestions for Authors
The manuscript submitted by Maes et al. presents a review of the role of PSMA in the biology of prostate cancer cells, and utilization of PSMA biomarker for targeted (endoradio)therapy and imaging. The manuscript is concise and clear. The review will be beneficial for both fundamental research and clinical communities.
Major
The authors are recommended to amend the manuscript with the information on how the search of the literature was performed and what criteria were used for selecting the publications to include into the manuscript. Thousands of publications on PSMA-targeting radiopharmaceuticals have been published, and only few were selected by the authors. The selection should be justified in the review. In that respect the title can then be more specific and limited to the impact of ADT on PSMA imaging and endoradiotherapy.
Minor
1. Abstract: Give the full name of ADT on the first appearance.
2. The radiopharmaceutical names should be corrected throughout the text according to the respective nomenclature rules (see Coenen et al., Nucl Med Biol, 2017, 55, v-xi).
3. The correct symbol for angstrom is Å.
4. Table 1. The parenthesis can be removed: (PSMA expression in prostate carcinoma).
Author Response
The authors kindly thank the reviewer for the time invested and for the suggestions made to improve the manuscript.
Below please find the replies to the comments made.
Major comments:
Criteria used for selecting the publications have been highlighted in bold in the adjusted manuscript at the end of the introduction as suggested..
Minor comments
-The full name of ADT was used on its first appearance in the abstract.
-The radiopharmaceutical names have been corrected according the nomenclature rules as per the paper by Coenen et al. (Nucl Med Biol 2017, 55, v-xi).
-The correct symbol for angström has now been implemented.
-The parenthesis in table 1 has been removed.
Reviewer 2 Report
Comments and Suggestions for Authors
Dear authors,
Thank you very much for allowing me to review your manuscript.
The present work describes the history, biology, pathophysiology, and imaging implications of PSMA for prostate cancer. The authors performed a thorough review and described their findings in detail, at times even erring on the side of too much information rather than getting their point across. Unfortunately, as is, this manuscript is neither innovative (as any other review) nor informative because it is almost impossible to retrieve the knowledge encoded in a hard-to-follow structure.
From my perspective, although the studies cited are relevant, the paper fails to deliver a coherent message, especially considering the discussion section, which is not opinionated and lacks direction. In order to properly inform readers, considering the broad audience of IJMS, I would advise the authors to revisit their discussion and try to format the content in a more front-loaded manner. Organizing paragraphs more logically, following a structure such as claim-data-warrant, would make this text more pleasing to read and informative. Structurally, I would also suggest revising the disposition of paragraphs to avoid long text blocks, such as the the 58-line paragraph that spans pages 5 and 6.
Ultimately, it would be beneficial to put yourselves in the shoes of readers who are not aware of the objective of this narrative review and revise it so that a previously uninformed person has a clear understanding of the current consensus on PSMA's role in prostate cancer diagnosis and treatment.
I commend the authors for their thorough research and hope that my comments help them improve its reach and appeal.
Specific points for improvement:
- Androgen Deprivation Therapy is not defined before being abbreviated in the abstract.
- The abstract and the second paragraph of the introduction are too similar, almost word for word. Therefore, the abstract fails to properly convey the topics discussed after the introduction.
- Capitalize "signaling" in the header of section 3.2
- Although the focus of the article is prostate carcinoma, it could be mentioned that PSMA is not exclusive to or pathognomonic of this type of cancer, and PSMA-ligand imaging has been recently detecting PSMA in extraprostatic tumors such as HCC
- "weeks" and "days" are used interchangeably with "w" and "d" in Table 2, use a consistent format throughout
Author Response
The authors kindly thank the reviewer for the time invested and for the suggestions made to improve the manuscript.
Below please find the replies to the comments made.
Major comments:
As suggested, the discussion section has been split up in different topics in order to properly inform the broad readership of IJMS. We hope that the discussion in its current form may provide a more clear understanding of the topic addressed (see headings added in bold in the discussion section).
Minor comments/specific points for improvement
-The full name of ADT is now used on its first appearance in the abstract (added in bold).
-The abstract (see text in bold) has been largely rewritten as to properly convey the topics discussed , in line with the modifications implemented in the discussion section.
-It is not clear to the authors where “signaling” should be capitalized, which section does the reviewer mean by section 3.2.
-PSMA expression in other types of tumors is not the topic of this manuscript and has been reviewed and reported more recently by our group.
-The authors understand that the use of days, weeks and months in table may be less coherent. However, these are the exact data referred to by the authors in their respective papers. The most straightforward approach would be the use of days. However, some authors have used months and different months include different number of days, so with what number should we than multiply the months? The authors prefer to leave it as such in order to not misrepresent data by other authors.